# Persistence of Anti-SARS-CoV-2 Antibodies in Non-Hospitalized COVID-19 Convalescent Health Care Workers

**DOI:** 10.3390/jcm9103188

**Published:** 2020-10-01

**Authors:** Margherita Bruni, Valentina Cecatiello, Angelica Diaz-Basabe, Georgia Lattanzi, Erika Mileti, Silvia Monzani, Laura Pirovano, Francesca Rizzelli, Clara Visintin, Giuseppina Bonizzi, Marco Giani, Marialuisa Lavitrano, Silvia Faravelli, Federico Forneris, Flavio Caprioli, Pier Giuseppe Pelicci, Gioacchino Natoli, Sebastiano Pasqualato, Marina Mapelli, Federica Facciotti

**Affiliations:** 1Department of Experimental Oncology, European Institute of Oncology IRCCS, via Adamello 16, 20139 Milan, Italy; margherita.bruni@ieo.it (M.B.); valentina.cecatiello@ieo.it (V.C.); angelica.diazbasabe@ieo.it (A.D.-B.); georgia.lattanzi@ieo.it (G.L.); erika.mileti@ieo.it (E.M.); silvia.monzani@ieo.it (S.M.); laura.pirovano@ieo.it (L.P.); francesca.rizzelli@ieo.it (F.R.); claraemilia.visintin@ieo.it (C.V.); giuseppina.bonizzi@ieo.it (G.B.); piergiuseppe.pelicci@ieo.it (P.G.P.); gioacchino.natoli@ieo.it (G.N.); sebastiano.pasqualato@ieo.it (S.P.); marina.mapelli@ieo.it (M.M.); 2Department of Oncology and Hemato-Oncology, University of Milan, via Festa del Perdono 7, 20122 Milan, Italy; 3School of Medicine and Surgery, University of Milano-Bicocca, via Cadore 48, 20900 Monza, Italy; marco.giani@unimib.it (M.G.); marialuisa.lavitrano@unimib.it (M.L.); 4The Armenise-Harvard Laboratory of Structural Biology, Department of Biology and Biotechnology “L. Spallanzani”, University of Pavia, via Ferrata 9, 27100 Pavia, Italy; silvia.faravelli1@gmail.com (S.F.); federico.forneris@unipv.it (F.F.); 5Gastroenterology and Endoscopy Unit, Fondazione IRCCS Ca’ Granda Ospedale Policlinico Milano, Via F. Sforza 35, 20135 Milan, Italy; flavio.caprioli@unimi.it; 6Department of Pathophysiology and Transplantation, University of Milan, Via F. Sforza 35, 20135 Milan, Italy

**Keywords:** COVID-19, SARS-CoV-2, anti-SARS-CoV-2 antibodies, pro-inflammatory mediators

## Abstract

Although antibody response to SARS-CoV-2 can be detected early during the infection, several outstanding questions remain to be addressed regarding the magnitude and persistence of antibody titer against different viral proteins and their correlation with the strength of the immune response. An ELISA assay has been developed by expressing and purifying the recombinant SARS-CoV-2 Spike Receptor Binding Domain (RBD), Soluble Ectodomain (Spike), and full length Nucleocapsid protein (N). Sera from healthcare workers affected by non-severe COVID-19 were longitudinally collected over four weeks, and compared to sera from patients hospitalized in Intensive Care Units (ICU) and SARS-CoV-2-negative subjects for the presence of IgM, IgG and IgA antibodies as well as soluble pro-inflammatory mediators in the sera. Non-hospitalized subjects showed lower antibody titers and blood pro-inflammatory cytokine profiles as compared to patients in Intensive Care Units (ICU), irrespective of the antibodies tested. Noteworthy, in non-severe COVID-19 infections, antibody titers against RBD and Spike, but not against the N protein, as well as pro-inflammatory cytokines decreased within a month after viral clearance. Thus, rapid decline in antibody titers and in pro-inflammatory cytokines may be a common feature of non-severe SARS-CoV-2 infection, suggesting that antibody-mediated protection against re-infection with SARS-CoV-2 is of short duration. These results suggest caution in using serological testing to estimate the prevalence of SARS-CoV-2 infection in the general population.

## 1. Introduction

The Coronavirus disease-19 (COVID-19) is a respiratory illness caused by the Severe Acute Respiratory Syndrome Coronavirus 2 (SARS-CoV-2), a novel beta-coronavirus firstly described in Wuhan city, China, on December 2019 [1]. SARS-CoV-2 spreading has been declared pandemic in mid-March 2020 by WHO [2]. At present, the virus has infected more than 33 million people worldwide with an associated case mortality rate of 1% to 15%, depending on the country [3]. 

COVID-19 is associated with a broad range of mild-to-severe symptoms, potentially leading to hospitalization in Intensive Care Units (ICU) for the most severe cases. The respiratory tract is initially involved with possible development of severe interstitial pneumonia [4,5], albeit the gastrointestinal tract can also significantly participate in disease pathogenesis as a consequence of the expression of the ACE2 receptor, that mediates SARS-CoV-2 viral entry [6], on both alveolar and enteric epithelial cells [7]. Infected subjects manifest a complex clinical pattern appearing as early as two days post exposure and lasting several weeks [1].

Infection with SARS-CoV-2 induces a prompt activation of the immune system, finalized to the clearance of infected cells [8]. Innate and adaptive immune cells accumulate at the site of infection, where production of cytokines and inflammatory mediators may result in patient recovery or, in case of ineffective viral clearance, in hyperactivation of the immune system and development of severe complications, such as acute respiratory distress syndrome ARDS [4,9]. Overexpression of pro-inflammatory cytokines (i.e., IL-1 beta, IL-2, IL-6, IL-17, TNFα etc.) and impairment of humoral immunity have been described in patients with the most severe form of the disease [5]. Antibodies against SARS-CoV-2 proteins are produced as a consequence of the activation of the humoral arm of the immune system. Virus-specific IgM antibodies are secreted as first class of immunoglobulins, followed by the more specific IgG [10]. Among the latter, those specific for the viral Spike Receptor Binding Domain (RBD) when expressed at higher titer manifest direct neutralizing activity towards the viral entry into cells, as they prevent effective engagement of surface ACE2 receptors by the Spike protein [11,12]. The IgA response against SARS-CoV-2 has been shown to be rapid and persistent [13,14] and is associated with mucosal responses, including respiratory and gastrointestinal responses.

Serological testing is a valuable tool to monitor viral spreading throughout the population [15]. Furthermore, serological assays allow the identification of past infection in individuals with viral RNA levels undetectable by RT-PCR for epidemiological purposes [16]. Various commercial and in-house assays that utilize distinct viral antigens and detect different antibody classes are currently available. However, SARS-CoV-2 serological tests available on the market do not always allow systematic simultaneous detection of a wide antibody spectrum for several antigens in a reliable manner, and this may hamper a proper population testing for clinical or epidemiological purposes [17]. Conversely, serological enzyme-linked immunosorbent assays (ELISA) to detect immunoglobulins raised against the viral Spike Soluble Ectodomain (Spike) and its highly immunogenic receptor binding domain (RBD), or against the Nucleocapsid protein (N), provide promising results in terms of accuracy and reproducibility [11]. Recently, these ELISA assays have been used to show that neutralizing antibodies (nAb) against different viral antigens may decline after 20–30 days post symptoms onset, and that the magnitude of nAb response may be associated with disease severity in COVID-19 patients [18].

In order to measure the presence and variation of antibody responses against different viral proteins, we set up and validated an in-house direct ELISA assay based on three distinct SARS-CoV-2 viral antigens, i.e., eukaryotically-expressed RBD and Spike and bacterially-expressed Nucleocapsid protein. Using this assay, we simultaneously measured IgM, IgG and IgA anti-viral antibodies titers in the sera of COVID-19 patients, as well as levels of pro-inflammatory cytokines. In addition, we longitudinally collected the sera of 16 convalescent healthcare workers who tested positive for SARS-CoV-2 by nasopharyngeal (NF) swabs, and were symptomatic but not hospitalized. Our data show that humoral immune responses against SARS-CoV-2 correlated with disease severity in terms of both antibody titers, persistence over time and serum levels of pro-inflammatory cytokines. Notably, 90% of COVID-19 mildly symptomatic patients halved their anti-RBD IgG titers after 4 weeks from viral negativization, thus confirming the short lifespan of humoral immune responses against SARS-CoV-2.

## 2. Experimental Section

### 2.1. Human Subjects

Health care workers of two different COVID Hospitals in Milan (*n* = 16) with documented COVID-19 infection (by NF swab), not hospitalized but with manifested COVID-19 symptoms (Appendix A) were monitored for seroconversion by IgM, IgG and IgA serum levels at two time points after viral clearance between April and June 2020. The study has been conducted in accordance with the Standards of Good Clinical Practice, with the ethical principles deriving from the Helsinki Declaration and the current legislation on observational studies. Clearance from the Ethical Committee has been obtained (IEO1271). Additional study populations were ICU hospitalized severe COVID-19 patients (*n* = 24) and (*n* = 58) COVID-19 negative subjects whose sera were collected between April and June 2020. In total, 436 pre-COVID subjects enrolled in IEO studies between 2009 and 2015 were used to calculate the ROC curves for the assays.

The exclusion criterion was, for all subjects involved in the study, the inability to provide informed consent. The inclusion criteria were, for those not hospitalized with COVID-19, (i) being health care workers (medical doctors, practitioners, post-graduate students, nurses), potentially exposed to SARS-CoV-2 between February and June 2020, (ii) documented SARS-CoV-2 infection by NF swab, (iii) not being hospitalized for COVID-19; for those hospitalized with COVID-19: (i) documented SARS-CoV-2 infection by NF swab, (ii) being admitted in the ICU between February and June 2020 for COVID-19; for negative controls: (i) sera being collected before 2019.

### 2.2. Antigen Proteins Production 

The recombinant Spike SARS-CoV-2 glycoprotein receptor binding domain (RBD) and the soluble full-length trimeric ectodomain have been produced in mammalian HEK293F cells as glycosylated proteins by transient transfection with pCAGGS vectors generated in Prof. Krammer’s laboratory [11]. The constructs were synthesized using the genomic sequence of the isolated virus, Wuhan-Hi-1 released in January 2020, and contain codons optimized for expression in mammalian cells.

Briefly, HEK293F cells were seeded at a final concentration of 0.5 million/mL in freestyle medium (Thermo Fisher Scientific, Milano, Italy), incubated at 37 °C, 5% CO_2_ at 120 rpm O/N in an Eppendorf New Brunswick s41i incubator. The day after HEK293F cells were transfected using 1 µg of DNA per 1 × 10^6^ cells and a DNA: PEI MAX ratio of 1:5 in OptiMEM medium. Four hours post-transfection, the medium was supplemented with Peptone Primatone RL (Merck) to a final concentration of 0.6% *w*/*v*. Cells were then incubated for 6 days, checking cell viability daily if needed (a mortality higher than 30% is indicative of a toxic protein).

For protein purification, the culture supernatant was transferred to conical centrifuge tubes, cleared by centrifugation at 1000×·*g* for 15 min and filtered with 0.22 µm Stericup Filters. The filtered medium addition with 1:10 volume of 570 mM NaH_2_PO_4_ pH 8.0, 300 mM NaCl and loaded on a HisPrep Fast Flow 16/60 column (GE-Healthcare) equilibrated in 57 mM NaH_2_PO_4_, 300 mM NaCl. His-tagged protein was eluted with step gradients of 50-100-150-235 mM imidazole. Peak fractions were pooled, dialyzed overnight against PBS and concentrated to 0.4 mg/mL (Spike soluble) or 1.0 mg/mL (RBD) in 10 kDa-MWCO Amicon filter units. 

Retrieved proteins were quantified, flash frozen in liquid nitrogen in aliquots and stored at −80 °C.

His-tagged SARS-CoV-2 full length N-protein plasmid (kind gift of David D. Ho, MD, Columbia University, New York, NY, USA) was transformed in E.coli BL21 pLysS cells. Protein expression was induced with 0.5 mM IPTG and carried on at 18 °C overnight. Cells were harvested by centrifugation in lysis buffer (25 mM Tris-HCl pH 8.0, 500 mM NaCl, 1 mM DTT, 5% glycerol, 10 mM imidazole, with Calbiochem protease inhibitor Cocktail III). All following steps were carried out at 4 °C or using ice-cold buffers. Cells were lysed by sonication; lysate was cleared by centrifugation at 20,000× *g* for 40 min, then PEI (pH 7.5—final concentration 0.02%) was dropwise added, under stirring, and lysate was then further cleared by centrifugation at 20,000× *g* for 40 min. Next, 5 mL Ni-NTA beads per liter of culture, pre-equilibrated in lysis buffer, were added to the cleared lysate and protein binding was continued for 1 h in gentle agitation at 4 °C. Beads were washed with at least 20 column volumes of 25 mM Tris-HCl pH 7.4, 500 mM NaCl, 1 mM DTT, 5% glycerol, 50 mM imidazole and His-tagged protein was eluted with 4 column volumes of 25 mM Tris-HCl pH 7.4, 200 mM NaCl, 1 mM DTT, 5% glycerol, 250 mM imidazole. The eluted fractions containing protein were diluted with heparin buffer (25 mM Tris-HCl pH 7.4, 1 mM DTT) to reach final a NaCl concentration of 0.1 M and were subsequently loaded onto a Hi-Trap Heparin HP column (GE Healthcare) equilibrated in 25 mM Tris-HCl pH, 50 mM NaCl, 1 mM DTT (buffer A). A linear gradient reaching 100% buffer B (25 mM Tris-HCl pH, 1 M NaCl, 1 mM DTT) in 30 column volumes was applied and fractions containing His-tagged N-protein were pooled, concentrated and loaded onto a Superdex 200 16/60 size exclusion chromatography. Fractions containing N-protein were pooled. A 4 L culture yielded 12.5 mL of 1.12 mg/mL pure N-protein, which was flash frozen in liquid nitrogen in aliquots and stored at stored at −80 °C.

### 2.3. ELISA

The ELISA assay to detect immunoglobulins (Ig) uses fragments of the SARS-CoV-2 Spike glycoprotein (S-protein) and the Nucleocapsid (N) as antigens based on the protocol published in [11,19]. After binding of the proteins to a Nunc Maxisorp ELISA plate, and blocking aspecific bindings with PBS-BSA 3%, patients’ sera to be analyzed were applied to the plate to allow antibody binding at a final dilution of 1:200, revealed with secondary anti-human-IgG (BD, clone G18-145), IgM (Merck, Polyclonal code A6907), IgA (Biolegend, Poly24110) antibody conjugated to HRP. Samples are read on a Glomax reader at 450 nm. This ELISA test is not intended for commercial use and is currently under evaluation at the Italy’s Ministry of Health (Aut.Min.Rich. 15.05.2020) for emergency use approval. The assay has been validated with a cohort of *n* = 56 COVID-19 subjects (severe, moderate and mild disease) and *n* = 463 (subjects collected in pre-COVID era (between 2012 and 2015)). ROC curves have been implemented to determine the sensitivity and specificity of the assay (Appendix A).

### 2.4. Multiplexing Analysis of Sera Cytokines

Quantification of soluble biomarkers was performed in sera of patients collected immediately after virus clearance (2 consecutive negative NF swabs) and one month post virus clearance using a Luminex Immunoassay (Human Cytokine/Chemokine/GF ProcartaPlex 45plex, Thermo Fisher) with MAP technology according to manufacturer’s protocol. Samples were acquired on a Luminex 200SD and analyzed with Xponent software 4.2. The sera of healthy subjects (*n* = 20) collected between April and June 2020 as well as ICU COVID-19 patients (*n* = 24) were used as control groups.

### 2.5. Statistical Analysis and Sample Size

The categorical variables were described as absolute frequency and percentage. The continuous variables with normal distribution were described as median ± standard deviation (SD), whereas the continuous variables without normal distribution were given as median and range. Normality of continuous variables was checked with D’Agostino–Pearson omnibus normality test. The Mann–Whitney test or Student’s *t*-test for continuous variables, and the Chi-square or Fisher’s exact tests for categorical variables, were used to associate clinical variables with the result of SARS-CoV-2 serological test (positive or negative). The *p* values lower than 0.05, two-tailed, will be considered statistically significant. GraphPad Prism software was used for all statistical analyses. 

## 3. Results

### 3.1. Set up and Validation of the ELISA Assay

To evaluate the antibody response of individuals infected by SARS-CoV-2, ELISA assays were developed in-house by producing and purifying recombinant RBD, Spike and Nucleocapsid proteins of the SARS-CoV-2 virus following the protocols published in [19] (Figure 1A). 

The performances of these ELISA assays were assessed for the different viral antigens and classes of antibodies by determining ROC curves using (i) a cohort of 56 sera from COVID-19 patients collected between April and June 2020 and tested positive for nasopharyngeal swabs, and (ii) 436 pre-COVID-19 sera, collected between 2012 and 2015 (Appendix A). Anti-SARS-CoV-2 IgG showed the highest specificity and sensitivity, irrespective of the antigen used (Appendix A). Anti-RBD IgG showed a specificity and sensitivity of 97% and 95%, respectively, while the assay performed with the Spike ectodomain reached values of 98.5% and 77% and the one with the N protein values of 91% and 95% (Appendix A). These performances are in line with those published for both in-house and commercial assays approved for emergency use by the FDA [20,21]. The performance of IgA detection was high for the RBD assay (91.5% specificity and 95% sensitivity), while it was slightly lower for the N protein (85% and 69%) and for the Spike (73% and 71%). The performance of the IgM assay was comparatively lower for all the viral proteins tested (Appendix A).

The validated ELISA assays were then used to systematically test the antibody titers of different classes of SARS-CoV-2 specific antibodies in sera from the following groups of patients: (i) 24 severe COVID-19 patients admitted to ICUs; (ii) 16 health care workers from two hospitals in Milan, exposed to the virus between February and March 2020 and confirmed positive to SARS-CoV-2 RNA by RT-qPCR on nasopharyngeal swabs. Fifty-eight SARS-CoV-2-negative subjects collected between April and June 2020 were used as negative controls (Appendix A). Sera of the 16 health care workers were collected in the convalescence phase of the disease after two consecutively negative nasopharyngeal swab tests. Time between the first detection of the virus and the first negative swab ranged from 14 to 35 days from onset of symptoms to disappearance of viral RNA (Appendix A). These subjects all manifested clinical symptoms strongly related to SARS-CoV-2 infection, including fever, ageusia, anosmia, fatigue, myalgia, diarrhea, coryza and cough [5]. Two of them manifested a more severe disease course with episodes of dyspnea. None of the patients required hospitalization and they all recovered from the disease (Appendix A). 

### 3.2. Mild COVID-19+ Patients Manifest a Lower Antibody Titer as Compared to Severe Patients

Non-hospitalized COVID-19 subjects manifested a lower antibody titer as compared to severe ICU patients for all the tested antibody classes and viral antigens (Figure 1B–D). This finding is in accordance to what published for asymptomatic [22] and paucisymptomatic [14] patients whose antibody titers were detected using commercial ELISA or chemiluminescence assays against either the Spike or the N-protein. When comparing the presence of the different classes of antibodies, all the COVID-19 positive subjects resulted positive for the presence of IgG antibodies against all the viral antigens tested (Figure 1E). Interestingly, a few of them were IgM negative or with an antibody concentration close to the detection limit of the Spike and RBD assay, as compared to the N protein.

The observation that all of them instead showed N-specific IgM antibodies may be a genuine persistence of anti-N protein IgM or the consequence of a lower specificity of the N assay, possibly reflecting the high conservation of the N proteins among beta-coronaviruses other than SARS-CoV-2 [23]. Interestingly, 25% of the non-hospitalized COVID-19 patients did not develop RBD-specific IgA, and only 1 out of 4 developed N-specific IgA antibodies, a percentage that was instead above 80% for the hospitalized ones (Figure 1E). 

### 3.3. Mild COVID-19 Patients Show a Reduced Release of Pro-Inflammatory Cytokines

Since severe COVID-19 is associated with a strong release of pro-inflammatory cytokines [8], the sera from COVID-19 patients were analyzed for the presence of pro- and anti-inflammatory cytokines, chemokines and growth factors by multidimensional analysis (Figure 2, Appendix A). ICU patients, whose sera were collected in the acute phase of the disease, showed a sustained production of pro-inflammatory mediators, among which IL-6, IL-17A, IL-12p70, IL-1beta, IL-4, IL-5 and IL-13, all associated with the “cytokine storm” observed in very severe COVID-19 patients, were the most abundantly detected (Figure 2A). On the contrary, even in the early convalescent phase, those cytokines were undetectable in the sera of non-hospitalized COVID-19 patients (Figure 2A). Interestingly, pro-inflammatory cytokines—such as IFN-gamma, TNF, IL-23, IL-15, IL-21 and IP-10/CXCL-10—were detected both in the sera of severe ICU hospitalized and of non-hospitalized COVID-19 patients (Figure 2B). To note, chemokines involved in the recruitment in inflamed tissues of both monocytes and T cells like MCP1/CCL2, RANTES/CCL5, MIP1alpha/CCL3 and EOTAXIN/CCL11 (Figure 2C) were present at comparable concentrations in severe ICU hospitalized and in non-hospitalized patients, indicating active recruitment of immune cell populations also in milder forms of COVID-19. 

### 3.4. Kinetic of Antibody Persistence in Mild COVID-19 Patients

In order to evaluate the kinetics of antibody titers in convalescent non-hospitalized COVID-19 patients, serum Ig levels were measured at different time points, i.e., two days (T1) and one month (T2) after the first negative NF swab (Figure 3A). Interestingly, within a month after negativization of the viral RNA, RBD- and Spike-specific antibody titers halved in the sera of the vast majority of convalescent COVID-19 patients (Figure 3B,C). When tested against the RBD, 14/16, 13/16 and 14/16 patients showed a decrease in the antibody title ranging from 30% to 90% in their viral-specific IgM, IgG and IgA antibodies classes (Figure 3B,E). Similarly, 8/16, 11/16 and 13/16 patients showed a decrease of at least 50% of their Spike IgM, IgG and IgA antibody titers (Figure 3C,E). In both cases antibodies titers were still above the OD detection threshold. On the contrary, antibodies against the viral Nucleocapsid protein did not show a significant decrease at the second time point of evaluation (Figure 3D,E). 

Interestingly, similarly to the antibody titers, the presence of proinflammatory mediators in the sera of convalescent patients also decreased over time and became almost undetectable one month after a negative PCR for viral RNA, a finding that mirrors the successful control of the infection and the consequent switch off of the immune response (Figure 3F, Appendix A).

## 4. Discussion

During the last months many key aspects of the immune response to SARS-CoV-2 have been elucidated. However, given the complexity and diversity of the clinical manifestation of COVID-19 disease, several outstanding questions remain still to be addressed. 

Here we show that humoral immune responses against SARS-CoV-2 correlated with disease severity in terms of both antibody titers, persistence over time and serum levels of pro-inflammatory mediators. Moreover, we showed that the vast majority of COVID-19 mildly symptomatic patients analyzed in the study halved their anti-RBD antibody titers after 4 weeks from viral negativization, thus confirming the short lifespan of humoral immune responses against SARS-CoV-2.

Humoral immune response against SARS-CoV-2 proteins leads the production of antibodies against the portions of the viral proteins [10,11,12]. In this sense, serological tests, based on the search of specific anti-SARS-CoV-2 antibodies, represent a useful tool aimed at identifying patients who contracted the infection and, consequently, comparing the clinical course and eventual complications between the general population and population at risk, such as health care workers [15]. Importantly, measurable variations in the humoral response might account for a re-activation of the immune system as a consequence of viral re-exposure, both in healthcare workers and in the general population. Serological monitoring of antibody levels can thus provide information on the actual circulation of the virus, which can be used by decision makers to adapt safety and restriction measures according to the real presence of the virus within the population.

Nonetheless, the specificity and sensitivity of the different assays greatly vary among kits taking into consideration the different techniques implemented (ELISA, CLIA, lateral flow) and the antigens used (Spike ectodomain, S1-S2 of the Spike, Spike RBD, Nucleocapsid). Thus, only highly sensitive tests can detect with high accuracy whether people, including mildly symptomatic or asymptomatic subjects, have specific anti-SARS-CoV-2 antibodies present in their blood. 

The test utilized in this study is a robust ELISA assay imported from the laboratory of Prof. Krammer at Mount Sinai, that has been approved for emergency use by the FDA [11,19]. We reproduced its excellent performance in our lab, that allowed us to detect a broad range of antibody levels, spanning form those measured in the blood of severe hospitalized patients and not hospitalized mild COVID-19+ individuals. The ELISA assay has been validated with a cohort of more than 500 positive and negative subjects, giving rise to extremely high performance values. Specificity and sensitivity of the ELISA assays were high for anti-RBD IgG and IgA (92–97%) and slightly lower for IgM and the Spike and N proteins (70–85%). These performances are in line with those published for both in-house and commercial assays [20,21]. For this reason, this test is also being currently evaluated by the Italy’s Istituto Superiore di Sanita’ (ISS) for its emergency use approval. 

One additional key strength of this assay as compared to other types of serological assays is its flexibility, i.e., the possibility to simultaneously assess different classes of antibodies against a broad panel of SARS-CoV-2 antigens within the same assay. Thus, this ELISA assay gave us a comprehensive understanding of the magnitude and persistence of antibody titer against different viral proteins and their correlation with the strength of the immune response, as measured by the serum levels of pro-inflammatory mediators. 

The presence of few false positives among the COVID-negative population tested with the viral nucleocapsid protein as compared to the RBD might be a consequence of a mistakenly detection of anti-N antibodies previously raised against common cold coronaviruses which cross-react with the SARS-CoV-2 nucleocapsid [23]. The nucleocapsid protein is the more conserved protein among different coronaviruses. It is possible to speculate that antibodies produced against previous common cold coronaviruses (and cross-reacting with the SARS-CoV-2 antigens) might still be present in the sera at high levels, and therefore be detectable. As a consequence, when analyzed longitudinally, we observed that only the antibodies specific to SARS-CoV-2 decline while those aspecific and possibly reacting to previous coronaviruses remain detectable at the same levels over time. A similar observation was recently published by a large longitudinal study [24]. Moreover, a recent paper evaluated the persistence of anti-N specific antibodies raised against four different common cold coronaviruses in a cohort of HIV+ individuals followed longitudinally for more than 10 years [25]. The study confirmed that N-specific antibodies undergo fluctuations in their detection levels as a consequence of seasonal re-infections with a kinetic of 6–12 months. Interestingly, the authors reported that 2 out of 10 patients (20% of the individuals enrolled in the study) showed cross-reactive antibodies against the viral N-proteins of the four viruses, and in one of them these cross-reactive antibodies persisted over the years.

The duration of circulating IgG antibodies is still unclear and might depend on several factors, including the type and extent of immune response elicited upon the encounter with the virus [17]. In this study, non-hospitalized subjects showed lower antibody titers and blood pro-inflammatory cytokine profiles compared to patients in Intensive Care Units (ICU), irrespective of the antibodies tested. This finding is in accordance to what published for asymptomatic [22] and paucisymptomatic [14] patients whose antibody titers were detected using commercial ELISA or chemiluminescence assays against either the Spike or the N-protein. 

Anti-RBD IgA antibodies manifested a similar kinetic compared to that of IgG. IgA response against SARS-CoV-2 has been reported to be rapid and persistent [18,19] and possibly associated with mucosal immune response in the gut and lungs. Notably, IgA production has been associated with disease severity, suggesting that IgA production might occur locally at the mucosal sites, possibly correlating with the viral load, the duration of the viral exposure and the virus entry route [13,26]. Consistently, a recent communication [14] confirmed that the highest levels of IgG and IgA antibodies against the Spike S1 domain, encompassing the N-terminal half of the protein with the RBD, were associated with severe disease [13,14].

Severe hospitalized COVID-19 patients overexpressed pro-inflammatory cytokines (i.e., IL-1 beta, IL-2, IL-6, IL-17, TNFα). In one of the very first reports of the clinical course of COVID-19 patients, as early as March 2020, serum increase in interleukin (IL)-2, IL-7, GMCSF, IP-10, MCP 1, MIP1-α, and TNF-α was associated to disease severity [5]. Elevated IL-6 levels were detected in hospitalized patients and have been associated with ICU admission, respiratory failure, and poor prognosis in several studies [5,27,28]. Presently, conflicting results regarding IL-1b and IL-4 have been reported [29,30,31]. The elevation of pro-inflammatory cytokines, albeit being widely described in COVID-19 patients, does not seem presently to have prognostic value, because they do not always differentiate moderate cases from severe cases [32]. Levels of IL-6 at first assessment might predict respiratory failure [33], other publications with longitudinal analyses demonstrated that IL-6 increases fairly late during the disease’s course, consequently compromising its prognostic value at earlier stages [34]. Moreover, serum concentrations of KL-6, a molecule elevated in serum of patients with interstitial lung diseases (ILDs), such as idiopathic pulmonary fibrosis and hypersensitivity pneumonitis, was recently proposed to be capable of differentiating between severe and mild COVID-19 patients, being mainly produced by damaged or regenerating alveolar type II pneumocytes [35,36].

Conversely, IP-10, MCP-3, and IL-1ra were capable of differentiating between severe and mild COVID-19 patients [32]. Interestingly, MIP 1 alpha, IL8 and Eotaxin, similarly to the results published by Long et al. [22], were expressed to a greater extent by healthy subjects compared to COVID-19 patients. Human MIP 1 alpha and Eotaxin were reported to be potent inhibitors of M-tropic HIV-1 infection, and were therefore considered as potential HIV-1 inhibitors [37]. A similar protective mechanism of action might be envisaged in SARS-CoV-2 infection. 

We also observed that during non-severe COVID-19 infections, pro-inflammatory cytokines are produced and correlate with the severity of the disease. Similarly to anti-SARS-CoV-2 antibodies, pro-inflammatory mediators also decreased within a month after viral clearance, as expected upon the resolution of the disease. 

Overall, we suggest that the decline in antibody titer and pro-inflammatory cytokines is a common characteristic of SARS-CoV-2 infection. This study therefore has important implications for the use of serological testing for the monitoring of infection outbreaks against re-infection with SARS-CoV-2. Our results indicate that the detection of antibodies with serological assays for epidemiological and monitoring purposes in non-hospitalized seroconverted COVID-19+ subjects, who most likely represent the majority of people who encountered the virus, is only highly reliable within a limited window of time after viral clearance.

## Figures and Tables

**Figure 1 jcm-09-03188-f001:**
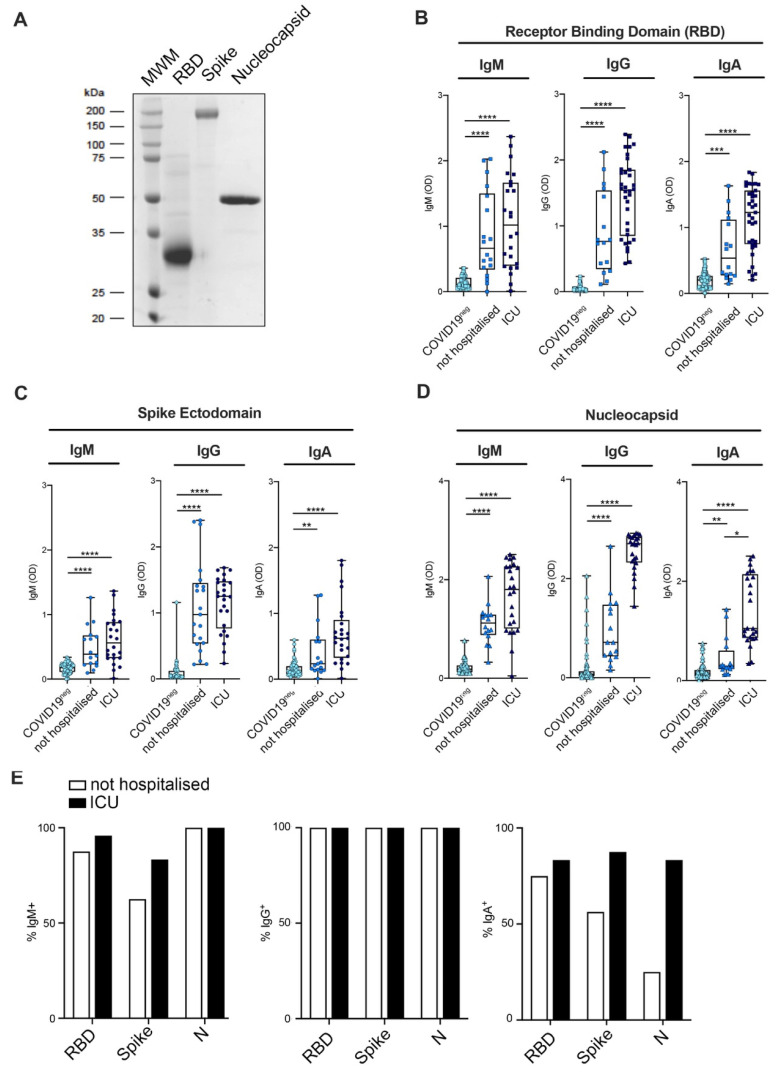
SARS-CoV-2 specific antibody levels of severe and mild COVID-19 patients. (**A**) Coomassie-stained SDS-PAGE showing receptor binding domain (RBD), Spike and Nucleocapsid (N) purified recombinant proteins used in the ELISA assays. (**B**–**D**) IgM, IgG and IgA levels in the sera of healthy subjects (light blue symbols, *n* = 58), non-hospitalized COVID-19 (blue symbols, *n* = 19) and intensive care unit (ICU) COVID-19 (dark blue symbols, *n* = 24) patients in ELISA assays against the RBD (**B**), the Spike ectodomain (**C**) and the N (**D**) SARS-CoV-2 viral proteins. (**E**) Differences in IgM, IgG and IgA antibody titers against RBD, Spike soluble and N protein in ICU (black bars) and non-hospitalized COVID-19 patients (white bars). *p* < 0.05 (*), *p* < 0.01 (**), *p* < 0.001 (***), *p* < 0.0001 (****) were regarded as statistically significant.

**Figure 2 jcm-09-03188-f002:**
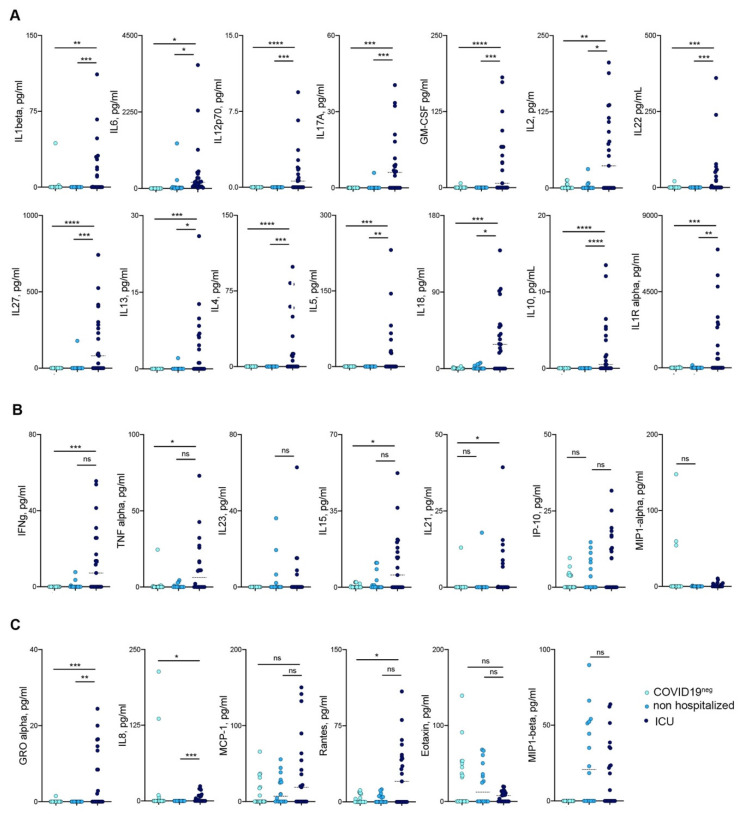
Cytokine levels in sera of COVID19 patients. (**A**) Cytokines significantly different between ICU (dark blue symbols) and non-hospitalized (blue symbols) COVID-19 patients. (**B**) Cytokines not significantly different between ICU (dark blue symbols) and non-hospitalized (blue symbols) COVID-19 patients (**C**) Chemokines levels in sera of patients (ICU, dark blue, not hospitalized blue symbols, healthy subjects light blue). *p* < 0.05 (*), *p* < 0.01 (**) *p* < 0.001 (***), *p* < 0.0001 (****) were regarded as statistically significant. ns, not significant.

**Figure 3 jcm-09-03188-f003:**
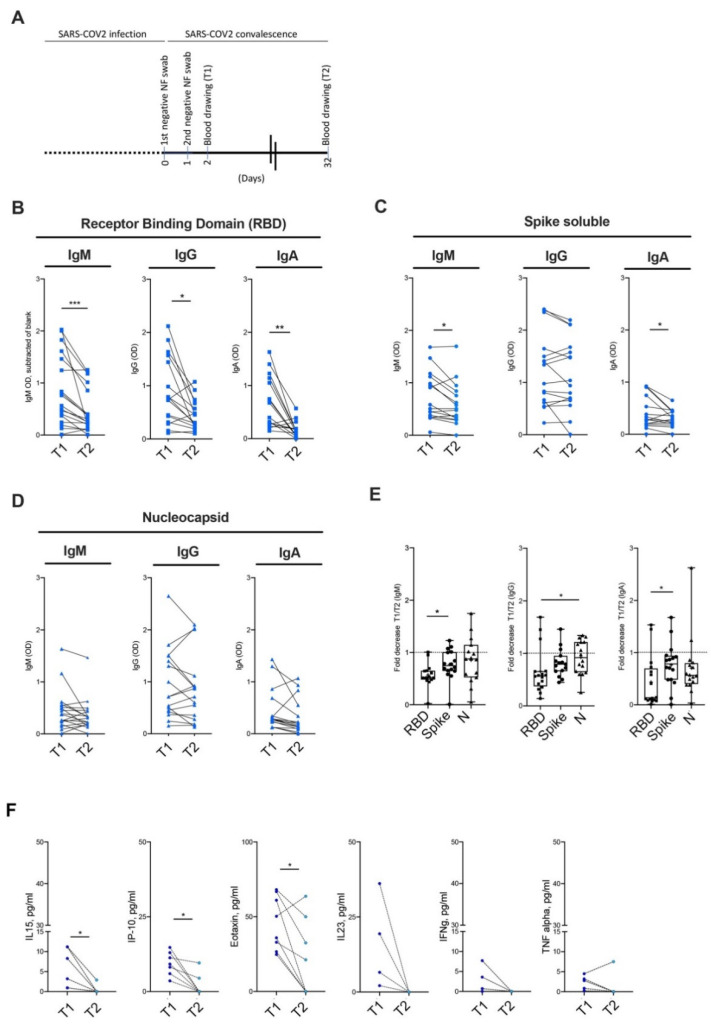
Longitudinal evaluation of antibody titers and cytokines in non-hospitalized COVID-19 patients (against all viral antigens). (**A**) Schematic representation of the study (**B**–**D**) IgM, IgG and IgA levels in the sera of non-hospitalized COVID-19 patients immediately (T1) and one month after (T2) cessation of viral detection by PCR, in the ELISA assays against the RBD (**B**), the Spike (**C**) and the N (**D**) SARS-CoV-2 viral proteins. (**E**) Cumulative fold decrease between T1 and T2 antibody titers in ELISA assays against the RBD (squares), the Spike ectodomain (circles) and the N (triangles) SARS-CoV-2 viral proteins. (**F**) Longitudinal variation of serum cytokines and chemokines in non-hospitalized COVID-19 patients. Statistical significance was calculated using Kruskal–Wallis nonparametric test for multiple comparisons. *p* < 0.05 (*), *p* < 0.01 (**) *p* < 0.001 (***) were regarded as statistically significant. ns, not significant.

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
