# Peer review of "Persistence of Anti-SARS-CoV-2 Antibodies in Non-Hospitalized COVID-19 Convalescent Health Care Workers"

_jcm, 2020, doi:10.3390/jcm9103188_

Round 1

Reviewer 1 Report

In this work, the authors describe that mildly symptomatic patients halved their antibody titer after 4 weeks. These results confirm the short lifespan of humoral immune response. Major comments: 1. Supplementary files are not available for downloading, yet they are cited in the manuscript, which is not ideal. 2. Even though the authors mention that antibody test could be useful for health care workers. Do the authors think that this test could be useful for general public as well?. I agree with the authors that antibody test are useful at identifying patients who contracted infection, to compare clinical course and eventual complications. However, is it really worthwhile spending resources to detect people that was infected, but still may get re-infected?. 3. It is mentioned along the lines in the discussion that common cold coronaviruses antibodies may be present long after infection, but not in SARS-CoV-2. It woud be interesting if the authors could expand their discussion on this. Minor comments: Line 39: ‘rapid’ Line 51: The correct term is mortality rate Line 87 evolution of antibody response? The word evolution is technically incorrect as only living organisms can evolve. Changes maybe? Figure 1B: The covid 19 negative patients are negative by PCR?

Author Response

Reviewer 1

We thank the reviewer for the comments that enabled us to improve the quality of our manuscript.

  1. According to the reviewer suggestion we changed the incorrect terminology in lines 39, 51, 87
  2. added a paragraph on the appropriateness of using serological tests to monitor the virus exposure in the general population (lines 309-314)
  • included a comment on the persistence of anti-N antibodies against common coronaviruses (lines 350-357)

Major comments:

  1. Supplementary files are not available for downloading, yet they are cited in the manuscript, which is not ideal.

We apologize, there must have been a problem with the uploading system. We provide the supplementary files for the reviewer.

  1. Even though the authors mention that antibody test could be useful for health care workers. Do the authors think that this test could be useful for general public as well?. I agree with the authors that antibody test are useful at identifying patients who contracted infection, to compare clinical course and eventual complications. However, is it really worthwhile spending resources to detect people that was infected, but still may get re-infected?.

We thank the reviewer for the comment. This is indeed an important point.

Molecular diagnostic tools detecting the actual presence of the SARS-CoV-2 virus allow the active surveillance of infected people. This tool is important both for healthcare workers and for the general population to efficiently contain the spreading of the virus.

Serological tests are instead valuable tests for epidemiological studies. In this context, variations in the serum levels of antibodies can provide insightful information on the viral re-exposure. Presently, it is still not clear if viral re-exposure equals re-infection. Measurable variations in the humoral response might account for a re-activation of the immune system as a consequence of a re-exposure to the virus. Importantly, the re-exposed subject might not be infectious (because of an efficient immune response) but serological monitoring of antibody levels in those individuals can provide information on the actual circulation of the virus, which can be used by decision makers to adapt safety and restriction measures according to the real presence of the virus within the population.

Whilst molecular testing can be performed only in dedicated laboratories with high biosafety levels, serological tests can be performed in any clinical or research laboratory by minimally trained technicians. From an economical point of view, molecular testing is way more expensive than serological testing. It has been reported that low income countries in the last months experienced difficulties in performing large scale molecular testing for SARS-CoV-2. In this scenario, serological tests helped governments to triage the population with relatively cheap systems, still collecting information on the presence of the virus that can be validated by molecular tests only in the subset of seroconverted subjects. An interesting report came from Peru at the peak of the pandemic in May. The Ministry of Health used rapid serological tests to prioritise the visits according to age, risk factors, and severity of symptoms. Rapid antibody tests were used to triage the population. A report indicated that among 355,604 people which had been triaged through this system, 42,534 tested positive by molecular testing (11.96%), 26,362 of whom (almost 62%, roughly 2 out of 3) had been detected by use of serological tests (Ministry of Health, Peru. COVID-19 in Peru. https://covid19.minsa.gob.pe/sala_situacional.asp).

We added this comment in the discussion in lines 310-315.

  1. It is mentioned along the lines in the discussion that common cold coronaviruses antibodies may be present long after infection, but not in SARS-CoV-2. It would be interesting if the authors could expand their discussion on this.

Our data indicate that RBD-specific antibodies against the SARS-CoV-2 virus are short-lived, starting to decline, at least in mild symptomatic patients, as early as 4 weeks post infection. The study of Shwan et al (REF 18 in the manuscript) also indicated 30 to 90 days post infection as the temporal window to detect anti-spike SARS-CoV-2 antibodies. Our study, in line with the recently published Scandinavian epidemiological study (REF 26 in the manuscript), indicates that N-specific antibodies might have a longer persistence. Similarly, a recent paper published while our manuscript was under revision (REF 1, Edridge et al, Nat Med published online 14 sept 2020) evaluated the persistence of anti-N specific antibodies raised against 4 different common cold coronaviruses in a cohort of HIV+ individuals followed longitudinally for more than 10 years. The study confirmed that N-specific antibodies undergo fluctuations in their detection levels as a consequence of seasonal re-infections with a kinetic of 6-12 months. Interestingly, the authors reported that 2 out of 10 patients (20% of the individuals enrolled in the study) showed cross-reactive antibodies against the viral N-proteins of the 4 viruses, and in one of them these cross-reactive antibodies persisted over the years.

This finding supports our hypothesis that the lower specificity of the ELISA test against the SARS-CoV-2 N protein and the enhanced persistence of anti-N antibodies compared to anti-RBD antibodies might be correlated  to the detection of previous common cold coronaviruses due to the high conservation of the N-protein among beta-coronaviruses.

We cited the results of this work in our manuscript (lines 350-357 of the revised version) commenting our results in line with these novel findings.

Minor comments:

Figure 1B: The covid 19 negative patients are negative by PCR?

COVID-19 sera belonging to otherwise healthy people that were collected between 2012 and 2015, thus by definition never exposed to SARS-CoV-2.

Reviewer 2 Report

Dear authors, In this study,  An ELISA assay containing recombinant SARS-CoV-2 Spike Receptor Binding Domain (RBD), Soluble Ectodomain (Spike), and full length nucleocapsid protein (N protein) has been investigated. Sera from healthcare workers affected by non-severe COVID-19, patients hospitalized in Intensive Care Units (ICU) and SARS-CoV-2-negative subjects were enrolled for the presence of IgM, IgG and IgA and pro-inflammatory cytokines. In non-severe COVID-19 infections, antibody titers against RBD and Spike, pro-inflammatory cytokines decreased within a month after viral clearance, uggesting that antibody-mediated protection against re-infection with SARS-CoV-2 is of short duration and the use of serological testing to estimate the prevalence of SARS-CoV-2 infection need caution.

MAJOR REVISIONS:

Human Subjects:

  • Can the authors better explain the selection of population? What are exclusion/inclusion criteria?
  • Can the authors listed if these patients are affected of some comorbidities that can modify the expression of cytokines? For example Diabetes.. etc I ask to authors adding these informations in Supplementary Table 1: Patients’ clinical characteristics
  • MIP 1 ALPHA, IL-8 and EOTAXIN, are the only investigated cytokines that resulted increase in COVID19 negative , although no statistical significant. Can the authors comments these results in the discussion? Can the authors believe that these markers can be used to differentiate the healthy from the patients?
  • Can the author provide some combination of markers that can increase the specificity and sensibility of each single marker?
  • “Severe hospitalized COVID-19 patients overexpressed pro-inflammatory cytokines (i.e. IL-1 348beta, IL-2, IL-6, IL-17, TNFα)” can the authors better explain these phrases?

MINOR POINT:

  • “The respiratory tract is initially involved with possible development of severe interstitial pneumonia”  

Can the authors better explain this point? I suggest to use this references:

d'Alessandro M, Cameli P, Refini RM, et al. Serum KL-6 concentrations as a novel biomarker of severe COVID-19 [published online ahead of print, 2020 May 29]. J Med Virol. 2020;10.1002/jmv.26087. doi:10.1002/jmv.26087

D'alessandro M, Bennett D, Montagnani F, et al. Peripheral lymphocyte subset monitoring in COVID19 patients: a prospective Italian real-life case series [published online ahead of print, 2020 May 14]. Minerva Med. 2020;10.23736/S0026-4806.20.06638-0. doi:10.23736/S0026-4806.20.06638-0

Author Response

We thank the reviewer for the insightful comments that we added to the manuscript and that contributed to its improvement.

  1. Can the authors better explain the selection of population? What are exclusion/inclusion criteria?

Mildly symptomatic COVID-19

_exclusion: inability to provide informed consent

_inclusion: documented SARS-CoV-2 infection by NF swab, being health care workers (medical doctors, practitioners, post-graduate students, nurses) potentially exposed to SARS-CoV-2 between February and June 2020

Severe:

_exclusion: inability to provide informed consent

_inclusion: documented SARS-CoV-2 infection by NF swab, being admitted in the ICU between February and June 2020 for SARS-CoV-2 infection

Negative:

_exclusion: inability to provide informed consent

_inclusion: sera collected before 2019.

We added this information in the revised material and methods

  1. Can the authors listed if these patients are affected of some comorbidities that can modify the expression of cytokines? For example Diabetes.. etc I ask to authors adding these informations in Supplementary Table 1: Patients’ clinical characteristics

We thank the reviewer for the comment. We added this information in the revised Supplementary Table 1

  1. MIP 1 ALPHA, IL-8 and EOTAXIN, are the only investigated cytokines that resulted increase in COVID19 negative , although no statistical significant. Can the authors comments these results in the discussion?

Can the authors believe that these markers can be used to differentiate the healthy from the patients?

Can the author provide some combination of markers that can increase the specificity and sensibility of each single marker?

We thank the reviewer for the possibility to comment on these points. Concerning the MIP 1 alpha, IL8 and Eotaxin expression by healthy subjects as compared to COVID-19 patients, the trend is not significant, possibly as a consequence of the relatively small number of subjects involved in the study. Interestingly, though, in the work of Long et al (REF 22 of the manuscript), these 3 molecules resulted also increased (not significantly) in healthy donors as compared to COVID-19 mild patients. Presently the significance of this finding is not known. Human MIP 1alpha and Eotaxin were reported to be potent inhibitors of M-tropic HIV-1 infection and were therefore considered as potential HIV-1 inhibitors (REF 10). A similar protective mechanism of action might be envisaged in SARS-CoV-2 infection. IL8 (CXCL8) is a potent attractor of neutrophils, which are key players in the initial phases of viral infections. An excess of neutrophil-lymphocytes ratio, though, has been indicated as a prognostic indicator of a more severe disease course in COVID-19. Thus, a fine balance of the concentrations of this chemokine might distinguish between a normal and a pathologic condition.

To reply to the other two related questions, we performed an unsupervised machine learning approach by random forest classifiers to test which markers more efficiently clustered COVID-19 versus healthy subjects.

As shown in Figure 1 for the reviewer, IL6, IL18 and MIP1b more significantly classified COVID-19 patients. Similarly, MIP 1 alpha and IL8 (but not Eotaxin) clustered more significantly healthy donors.

While there are reports that levels of IL-6 at first assessment might predict respiratory failure (REF 8), other publications with longitudinal analyses demonstrated that IL-6 increases fairly late during the disease’s course (REF 9). Conversely, IP-10, MCP-3, and IL-1ra were capable to differentiate severe versus mild COVID patients (REF 7)

Therefore, it is possible to suggest that the combinations “IL6, IL18, MIP1b” and “MIP1alpha, IL8” could eventually be used to differentiate healthy form COVID-19 patients better than the single markers alone. A larger cohort of negative and positive subjects should be however used to confirm these data.

We added these comments in the revised version of the manuscript.

Figure 1 for reviewer: Machine learning analysis based upon random forest approach to classify COVID 19 versus healthy subjects based upon serum biomarkers detection

  1. “Severe hospitalized COVID-19 patients overexpressed pro-inflammatory cytokines (i.e. IL-1 348beta, IL-2, IL-6, IL-17, TNFα)” can the authors better explain these phrases?

In one of the very first report of the clinical course of COVID-19 patients, as early as March 2020, serum increase of interleukin IL-2, IL-7, GMCSF, IP-10, MCP 1, MIP1-α, and TNF-α was associated to disease severity (REF 5 of the manuscript). Elevated IL-6 levels were detected in hospitalized patients and in several studies have been associated with ICU admission, respiratory failure, and poor prognosis (REF 2,3 and 5 of the manuscript). Conflicting results regarding IL-1b and IL-4 have been reported (REF 4-6). Elevation of pro-inflammatory cytokines, albeit being widely described in COVID-19 patients, does not seem presently to  have prognostic value, because they do not always differentiate moderate cases from severe cases (REF 7).

MINOR POINT:

  • “The respiratory tract is initially involved with possible development of severe interstitial pneumonia”  

We expanded the concept by adding the suggested references in lines 381-385 (new refs 36,37)

  1. Edridge D, et al. Seasonal coronavirus protective immunity is short lasting. Nature Medicine, September 14 2020
  2. Chen X, et al. Detectable serum SARS-CoV-2 viral load (RNAaemia) is closely associated with drastically elevated interleukin 6 (IL-6) level in critically ill COVID-19 patients. Infect. Dis. Published online April17, 2020
  3. Liu et al The role of interleukin-6 in monitoring severe case of coronavirus disease 2019. EMBO Mol. Med. Published online May 19, 2020.
  4. Fu S et al Virologic and clinical characteristics for prognosis of severe COVID-19: a retrospective observational study in Wuhan, China. https://doi.org/10.1101/2020.04.03.20051763.
  5. Gong J et al Correlation Analysis Between Disease Severity and Inflammation-related Parameters in Patients with COVID-19 Pneumonia. https://doi.org/10.1101/2020.02.25.20025643.
  6. Wen et al Immune cell profiling of COVID-19 patients in the recovery stage by single-cell sequencing. Cell Discov. Published online May 4, 2020.
  7. Yang, Y., et al. Plasma IP-10 and MCP-3 levels are highly associated with disease severity and predict the progression of COVID-19. Allergy Clin. Immunol. Published online April 29, 2020.
  8. Herold, T., et al Elevated levels of interleukin-6 and CRP predict the need for mechanical ventilation in COVID-19. Allergy Clin. Immunol. Published online May 18, 2020.
  9. Zhou, F., et al. Clinical course and risk factors for mortality of adult inpatients with COVID-19 in Wuhan, China: a retrospective cohort study. Lancet 395, 1054–1062.
  10. Cocchi F, et al. Identification of RANTES, MIP-1α, and MIP-1β as the major HIV-suppressive factors produced by CD8+T cells. Science, 270 (1995), pp. 1811-1815).

Round 2

Reviewer 2 Report

  • well done!